# Community Perceptions and Knowledge of Modern Stormwater Treatment Assets

Hadi Zamanifard [1], Edward A. Morgan [1,*] and Wade L. Hadwen [2]

1   Cities Research Institute, Griffith University, Brisbane, QLD 4111, Australia; hadi.zamanifard@gmail.com
2   Australian, Rivers Institute, Griffith University, Brisbane, QLD 4111, Australia; w.hadwen@griffith.edu.au
*   Correspondence: ed.morgan@griffith.edu.au; Tel.: +61-(0)7-3735-9248

**Abstract:** Modern stormwater treatment assets are a form of water sensitive urban design (WSUD) features that aim to reduce the volumes of sediment, nutrients and gross pollutants discharged into receiving waterways. Local governments and developers in urban areas are installing and maintaining a large number of stormwater treatment assets, with the aim of improving urban runoff water quality. Many of these assets take up significant urban space and are highly visible and as a result, community acceptance is essential for effective WSUD design and implementation. However, community perceptions and knowledge about these assets have not been widely studied. This study used a survey to investigate community perceptions and knowledge about stormwater treatment assets in Brisbane, Australia. The results suggest that there is limited community knowledge of these assets, but that communities notice them and value their natural features when well-maintained. This study suggests that local governments may be able to better inform residents about the importance of these assets, and that designing for multiple purposes may improve community acceptance and support for the use of Council funds to maintain them.

**Keywords:** water sensitive urban design; stormwater management, integrate water management; urban design; green infrastructure; community perceptions





## 1. Introduction

Water quality in urban creeks and rivers is highly affected by stormwater flows. Urbanisation leads to decreased permeability of the landscape, resulting in elevated stormwater flows and elevated inputs of water, sediment, nutrients and gross pollutants into creeks and rivers [1–4] These changes in stream hydrology and water quality negatively affect stream biota and the ecological functioning of urban streams [5,6].

In response to concerns regarding urban stream health and the impacts of stormwater flows, policies have been developed to ensure that stormwater running off all new developments is appropriately managed. In Australia, the resultant approaches are often referred to as Water Sensitive Urban Design (WSUD), which is a land use planning and environmental engineering approach to managing water more holistically in urban settings [7–10]. The philosophy focuses on the need to create water sensitive cities [11,12]. A water sensitive approach is designed to slow the flow of stormwater, to help manage flood risk and facilitate water treatment through settlement and assimilation processes, prior to discharge into local streams and rivers [3,4,7,13–15], and encompasses low impact development, sponge cities and similar approaches [16–18]. The intended outcome of WSUD is reduced loads and reduced event magnitudes of sediment, nutrient and coarse pollutants entering local waterways. Common stormwater treatment assets include swales, bioretention basins, biofiltration devices and constructed wetlands (note that there are many variations of each type) [14,17,19–23].

WSUD features represent important 'green infrastructure' within urban areas, in that they harness ecosystem services to protect the quality of receiving waterways [13,23–27].

In addition to their stormwater and flood management roles, these aesthetically pleasing green assets can make new urban developments more appealing to prospective residents [7,28–30]. If properly understood, the multiple benefits of WSUD features could help to justify the ongoing costs and complexities of installing and maintaining them.

WSUD features have become increasingly part of water management and planning policy in rapidly developing urban areas [10,11,18]. For example, in the area where this study is conducted, Queensland, Australia, WSUD is now a legislated requirement for all new developments. This means that new urban developments must have a stormwater management plan and adopt features of WSUD to treat stormwater onsite. The state planning policy has set performance outcomes and design objectives for nutrient removal via stormwater management that provide minimum reduction requirements for total suspended solids, total phosphorus, total nitrogen and gross pollutants for WSUD stormwater assets [31]. While there are undoubtedly other benefits associated with stormwater treatment assets, including flood control, stormwater harvesting and biodiversity [10,17,23,26,27], the specific design objectives of interest in this study related to their role as treatment systems, which is the common focus of stormwater features in the US, Europe and Australia [17].

Whilst the rationale and legislation around WSUD is compelling, there remain many significant knowledge gaps around the performance and environmental outcomes of WSUD features post-implementation [6,19,22,26,32–36]. Indeed, WSUD performance is generally not monitored at all and expectations on sediment and nutrient reductions, or other performance measures, are generally taken as modelled, without the necessary ground-truthing, although more recent studies have begun to investigate in-situ performance [22,23,33,35,37].

Many local government authorities are now responsible for the maintenance and performance of hundreds, if not thousands, of complex WSUD assets. These features can be highly varied in design and objective, look very different and require very different maintenance regimes [33]. If these features fail to meet performance expectations or create new issues that need management interventions, then WSUD risks losing the support of governments and the community as an effective approach to land-use planning and stormwater management [28,38]. As Sharma et al. [33] note, the effectiveness of WSUD features is a multifaceted issue, involving technical engineering performance, policy and public awareness and acceptance. Hence, community understanding and perception of WSUD stormwater assets is another vital element of ensuring that they are performing in a broader sense.

Community acceptance is well-established as an important element of water governance [39], most notably around alternative water sources [40,41]. Community engagement around water issues is widely recognised as a key part of improving water governance [39,41–44]. In the context of WSUD, designing systems that are acceptable to the community, while still achieving the desired outcomes, is vital [21,28,29,33,45]. In addition, it has been suggested that community support for WSUD is partly dependent on community knowledge of the purpose and function of these interventions, but public knowledge and awareness is often limited and hindered by jargon [46,47].

To build community-level support for the implementation of WSUD assets, it is necessary to understand how the community feels about these engineered systems and whether they understand their design objectives. Whilst there have been assessments of community knowledge in Australia about water and how communities engage with water [39,43,46,48], there have, to date, been few assessments of resident values and perceptions of WSUD or stormwater features anywhere in the world. Studies have tended to focus on municipalities perceptions [49] or proposing conceptual frameworks to understand perceptions [29,45]. Sharma et al. [33], used focus groups to investigate community perceptions of WSUD features in Adelaide, Australia. However, focus groups require significant time and resources and may not be practical to investigate perceptions of a large number of features. To address this knowledge gap, we conducted a pilot project to design and test a survey to demonstrate how resident surveys can be used to build increased understanding of the

multiple benefits and values attributed to WSUD assets, in this case, a single class of very common assets—bioretention basins.

## 2. Materials and Methods

### 2.1. Study Area

For this pilot study, the focus was on a single class of assets—bioretention basins—in a selected number of neighbourhoods in southern and western Brisbane (Figure 1). Brisbane has a sub-tropical climate, with hot, humid, wet summers and warm, dry winters. Monthly average temperatures range from around 30 °C in summer to 20 °C in winter, although summer temperatures can reach 40 °C [50]. Median monthly rainfall varies from around 120 mm in summer to 30 mm in winter. Importantly for the design of stormwater features, Brisbane experiences frequent heavy thunderstorms, especially in summer, with intense but relatively short periods of rainfall [50]. Brisbane's climate, like much of Australia, is strongly affected by the El Nino–La Nina cycles and can experience long droughts and severe flooding. Recently, for example, Brisbane experienced drought from 2000–2009, followed by flooding in 2010 and 2011 [51,52]. Climate change effects on the local climate are expected to exaggerate existing trends, with higher average temperatures and more extreme heat-waves predicted with high certainty, and more intense drought and heavy rainfall expected [53]. Stormwater features, especially bioretention basins, in these types of climate must be able to tolerate high rainfall intensity and remain effective after long-periods of very low rainfall [54,55]. This study was performed during a relatively typical summer that had seen some high intensity storm events, although surveys took place during a relatively long dry period.

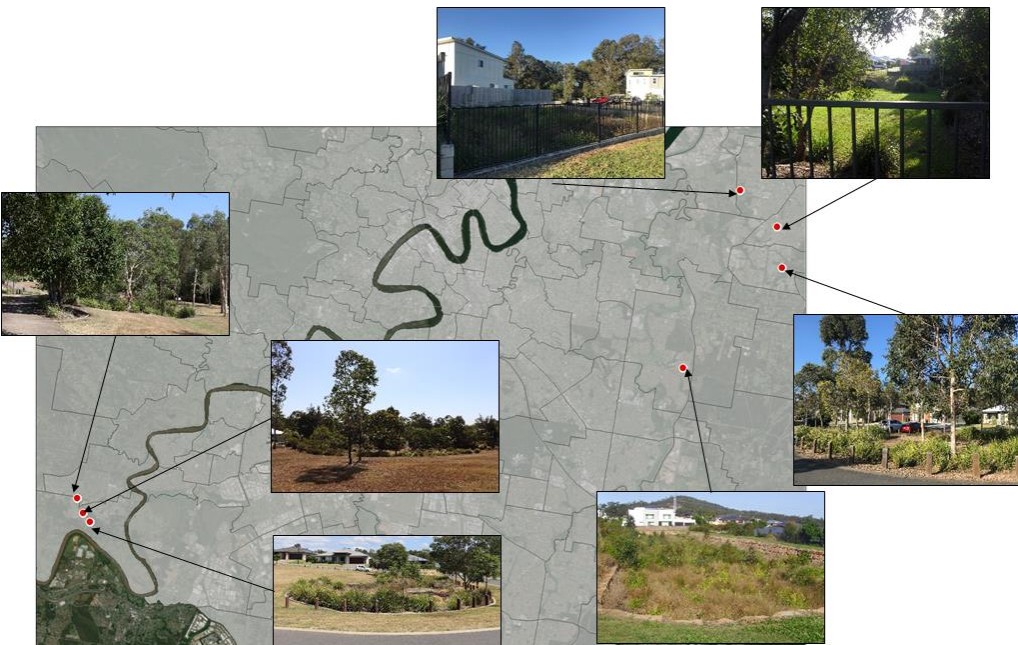

**Figure 1.** The seven case study bioretention basins selected in southern and western Brisbane.

Bioretention basins were selected as the focus of the study, as they are the most common class of WSUD assets in southeast Queensland. For the case study, candidate assets were selected on the basis of their size, age and condition assessment ratings, as determined using the stormwater asset database developed by the local council. In addition, site visits were conducted to consider the adjacent neighbourhood, network of paths and the likelihood that residents would drive or walk past the bioretention basin in question. Particular effort was taken to ensure that photos of the selected sites would be recognisable to residents, as we were keen to ensure that the respondents could confidently share their views of the system in question. Ultimately, from an initial selection of 15 potential biore-

tention basins, seven were selected for inclusion in the pilot survey covering a range of different designs and locations (Figure 1).

*2.2. Survey Development and Implementation*

To explore resident values and perceptions of bioretention basins in their local areas, we developed an online survey. The survey used a combination of closed and open-ended questions to ascertain resident knowledge and concerns (Supplementary Materials). Investigating resident perceptions using this mixed methods approach enables exploration of trends in the quantitative data, coupled with deeper insights afforded by the written responses in the form of qualitative data. The initial survey development revolved around key questions relating to the intent and purpose of the bioretention basins (without naming them) to specifically evaluate resident awareness and understanding of the system purpose and design. The survey was originally developed in-house at Griffith University, with questions and feedback from the local council team with responsibility for these features to improve and tailor the questions further.

Importantly, the challenge of avoiding technical terminology and jargon, common within the WSUD sector, was important. Previous research has shown a lack of public knowledge about the jargon used in a lot of water resource management [47]. Hence, the questions were designed to include multiple options of 'correct' answers to try and capture the 'layman's' terminology that people are more likely to recognise.

Although the survey was developed as an online survey, which makes it easy to complete on mobile devices and computers, getting to the target survey population represented a significant challenge. To try and reach as many residents as possible, we coupled site-based interception surveys, letterbox drops and Facebook group links to market the survey. To further encourage participation, we also provided the opportunity to win a cash prize as an incentive. To some extent encouraging participation was limited by the time and resources of the project. The aim was to test the methodology and understand challenges before attempting wider surveying of participants.

A significant challenge with surveying residents about these features is that they are very much 'local' features which are often hidden, landscaped and located at low points in the area. As a result, only people in a relatively small area are likely to have seen or interacted with any individual bioretention basin. Hence, separate surveys were required for each basin, with photos used to promote the survey locally. Given the different design features and different local communities targeted in this research, it was not possible to use other methods of increasing survey responses such as survey companies.

## 3. Results

*3.1. Demographic Details of Respondents*

A total of 31 completed surveys were returned across the seven case study areas. This represents a low response rate given the large number of flyers left in letterboxes, the site-based survey efforts and the Facebook-group advertisements deployed. The breakdown of respondent demographics is presented in Table 1.

The characteristics of the respondents was not a representation of Brisbane generally (Table 1). There was a high proportion of highly educated people, and a greater proportion of younger people. The higher proportion of the 35–44 age demographic may represent the group most likely to be engaged on Facebook, as well as those likely to buy a house in a newer greenfield development. The high education level probably reflects the nature of the subject of the survey.

The disproportionate number of respondents reporting 1–5 year residency likely reflects the nature of the surrounding housing, some of which will be relatively newly built. Indeed, these WSUD features are relatively new technologies, and the policies that determine targets are also relatively new. As a result, these features are more common in relatively newly built estates and precincts. It is important to note that as these features and communities age, perceptions may change and surveys may need to be repeated.

The demographics suggest that the survey respondents were likely 'self-selecting' based on their interest in and connection to the locally built environment and greenspace. This is supported by the fact that almost all of the respondents added written comments at the end of the survey. The results therefore reflect the views of the more engaged residents, rather than being representative of the population. Work by Dean et al. [39,43] investigating Australian residents knowledge and attitudes to water management identified five groups: 'disengaged', 'aware but inactive', 'active but not engaged', 'engaged but cautious' and 'highly engaged'. The demographic profile the survey respondents here aligns with Dean et al.'s 'engaged but cautious' and 'highly engaged' groups. We suggest, therefore, that the results are a first cut of the views of engaged residents (engaged on this specific issue). Notably, research has shown that these are a good group to target initially for engagement in water management issues, including WSUD [39,43,46,48].

**Table 1.** Demographic details of survey respondents, with comparisons (where relevant) to comparable data for the greater Brisbane region (Brisbane (%) from the Australian Bureau of Statistics (ABS)).

| | N (Total = 31) | % | Brisbane (%) [56] | | N (Total = 31) | % | Brisbane (%) [56] |
|---|---|---|---|---|---|---|---|
| **Gender** | | | | **Weekly Income** | | | |
| Male | 11 | 35 | 49.2 | $0–500 | 4 | 13 | 11.4 |
| Female | 16 | 52 | 50.8 | $501–1000 | 1 | 3 | 16.9 |
| Unspecified | 4 | 13 | — | $1001–1500 | 2 | 6 | 14.4 |
| **Age** | | | | $1501–2500 | 8 | 26 | 23.6 |
| 18–24 | 2 | 6 | 7.5 [a] | $2501–5000 | 3 | 10 | 19.9 |
| 25–34 | 5 | 16 | 15.1 | Greater than $5000 | 2 | 6 | 3.3 |
| 35–44 | 16 | 52 | 14.0 | Prefer not to say | 10 | 32 | 9.7 |
| 45–64 | 4 | 13 | 23.9 | **Residency Length** | | | |
| 65–74 | 2 | 6 | 7.8 | Less than a year | 3 | 10 | — [b] |
| 75+ | 1 | 3 | 5.6 | 1–5 years | 19 | 61 | — |
| **Education** | | | | 6–10 years | 4 | 13 | — |
| Secondary education | 2 | 6 | 17.8 | More than 10 years | 4 | 13 | — |
| Certificate or Diploma | 2 | 22 | 16.4 | | | | |
| Bachelor Degree | 8 | 26 | 22.9 | | | | |
| Graduate Diploma | 4 | 13 | 9.2 | | | | |
| Postgraduate | 9 | 29 | N/A | | | | |

[a] ABS data is divided into 15–19 and 20–24, the 20–24 percentage is used here as an approximation. [b] No data available for residency length.

### 3.2. Resident Perceptions of Uses of Bioretention Basins

In response to the question 'What do you think this space is for?', the answers 'stormwater treatment asset' or 'a natural filtration feature' covered almost half (48%) of responses (Figure 2), even though residents could pick more than one response. However, of these, only six respondents (19%) selected only one or both of the correct answers. Hence, most of those that recognised its primary 'as designed' function (to treat stormwater), also thought it had other functions.

The answer 'a flood control feature' appeared in 39% of responses suggesting that many respondents recognised that the asset had a water-related function, and 'greenspace' and 'conservation area' appeared in 48% of responses. Overall, residents generally recognised that the asset was purpose-built (or engineered) for a water or greenspace purpose, even if they were not sure of the exact use. As bioretention basins are often noticeable

within the landscape—several of those in this study were fenced off—it appears that residents often recognise it as a particular use of space, but rarely know what it is for.

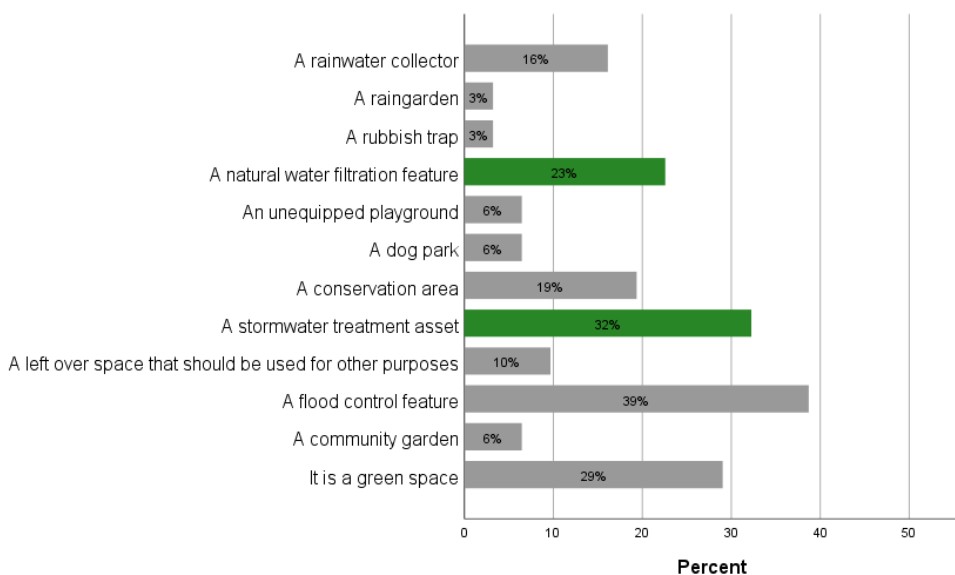

**Figure 2.** Respondent views of the purpose of the nominated bioretention basin in their local area. Respondents were able to select more than one answer.

For the 14 respondents that did include 'stormwater treatment asset' or 'a natural filtration feature', hence recognising its primary designed purpose, almost two-thirds (64%) didn't know what it was called, supporting our expectation that people wouldn't know the technical or jargon terms common within the WSUD space. Despite this, 57% of these 14 respondents felt that they had a very clear, clear or somewhat clear understanding of how the feature works to treat stormwater.

Many respondents provided comments reflecting a lack of knowledge and awareness of bioretention basins, highlighting the need for more resident education. Many statements reflected both their values and their lack of understanding, with many respondents indicating that they wished they knew more about these systems and querying why they had not been informed of the value and design intent of these systems. For example, one respondent discussing the function of the feature after being given information at the end of the survey noted '*But this was never explained to the residents—I now understand the environmental importance of the asset*'. Another resident offered an environmentally sound solution, by offering '*Educating the local neighbourhood through an email to residents in the local area (emails should be sent to reduce waste through using addresses on the electoral roll).*'

We did investigate whether there were significant differences in other responses between those who knew the primary purpose of the feature and those who did not. In most cases, there were no significant differences in the correlations of responses between each group, with the exception of desirability, and consequently the results are mainly discussed collectively in the sections below,

### 3.3. Resident Use of Bioretention Basins

While more than 80% of the respondents did not use the space occupied by the bioretention basin for any activity, those that did suggested that walking, relaxation, clean air and greenery were important activities for them in the vicinity of these assets (Figure 3).

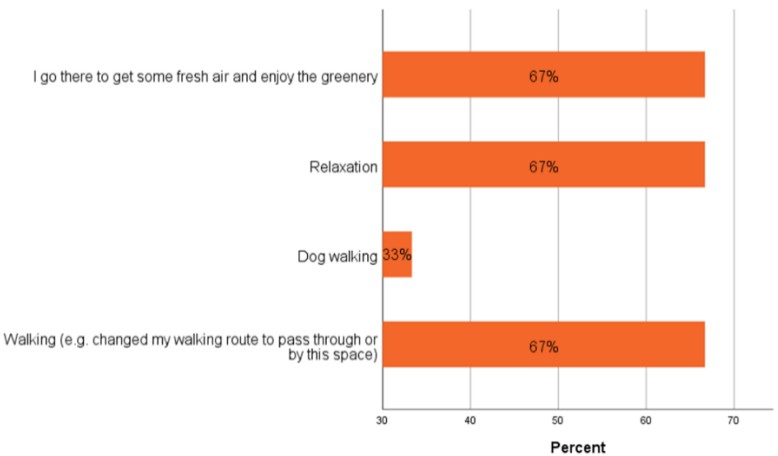

**Figure 3.** Respondent views on use of bioretention space in their local area.

*3.4. Resident Views and Perceptions of Benefits of Bioretention Basins*

Despite the lack of knowledge around the designed purpose and functioning of the bioretention basins, respondents did feel that there were benefits associated with these green assets (Figure 4). More than two-thirds of all respondents nominated bioretention basins as places with clear environmental benefits (providing space for local wildlife). Respondents also felt that bioretention basins could have social, aesthetic and health benefits. The strong environmental benefits were also borne out in respondents views on what they like about the bioretention basins (Figure 5)—the openness and greenery (50%), the wildlife (36%) and the trees (36%) were the dominant responses.

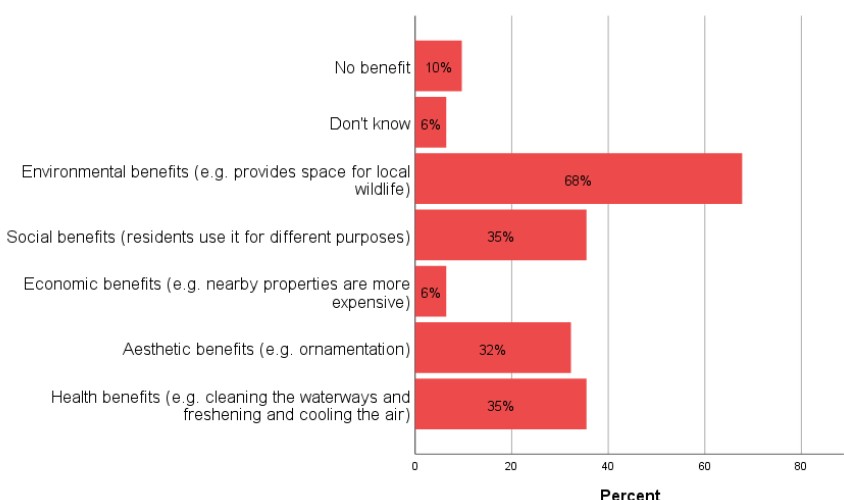

**Figure 4.** Respondent views of the benefits of their local bioretention basin.

It is clear that our survey respondents attributed a wide range of environmental values to their local bioretention basins. As noted by one respondent, '*It is nice to see green space for whatever reason, not being given the development green light. We have lost a lot of natural green space to developers in this area over last seven years.*' This comment highlights the fact that most WSUD systems have been installed relatively recently and in areas that are rapidly developing, so efforts taken to safeguard some green areas are often broadly supported by the community.

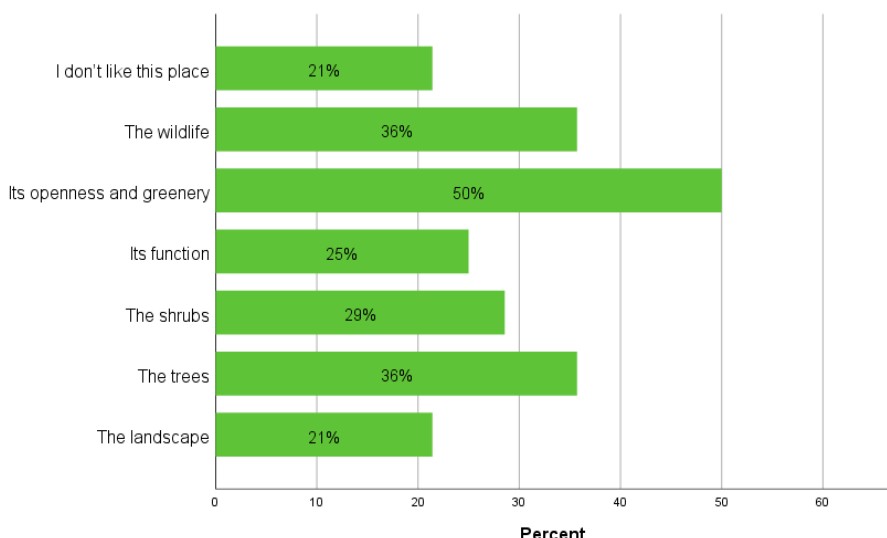

**Figure 5.** Respondent views on aspects that they like regarding their local bioretention basin.

### 3.5. Resident Views on Bioretention Basins

Even respondents with no knowledge of the purpose of bioretention basins tended to value having these systems in their local area (Figure 6). More than half of all respondents found them to be very desirable features in their local area and just one quarter felt that they were not desirable (Figure 6). Most (59%) were also happy with the level of maintenance of the green space, with less than one third of all respondents indicating that they were dissatisfied with current maintenance practices. Notably, a positive view on the feature was much more common among those that knew the primary function of the feature as a stormwater treatment asset/natural water filtration feature with all but one of that group (92%) selecting very desirable, desirable or somewhat desirable. The correlation was not statistically significant, however, likely due to the low numbers of respondents. Nonetheless, this does provide some evidence that greater awareness of the purpose of the feature resulted in a more positive view.

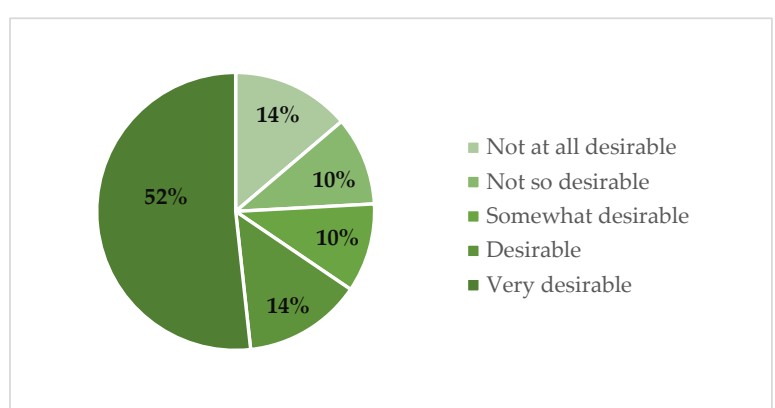

**Figure 6.** Respondent views regarding the desirability of their local bioretention basin.

While 30% of respondents liked everything about their local bioretention basin, more than half of the respondents (56%) indicated that poor maintenance was something they did not like about the space (Figure 7). Indeed, maintenance (48%) and better landscaping (55%) dominated responses for how the space could be improved, again suggesting that the values of local residents were more focused on weed maintenance and greenness/habitat values than on the designed functioning of the stormwater treatment asset.

The environmental viewpoints of respondents also came through the free text responses, with some residents concerned about maintenance regimes. As noted by one

concerned resident, '*Council always spray the weeds with something and I don't think that is good for the waterways*'. For a system designed to protect environmental health, this statement raises questions around whether the maintenance regime fits in with the broader approach to protecting the environment.

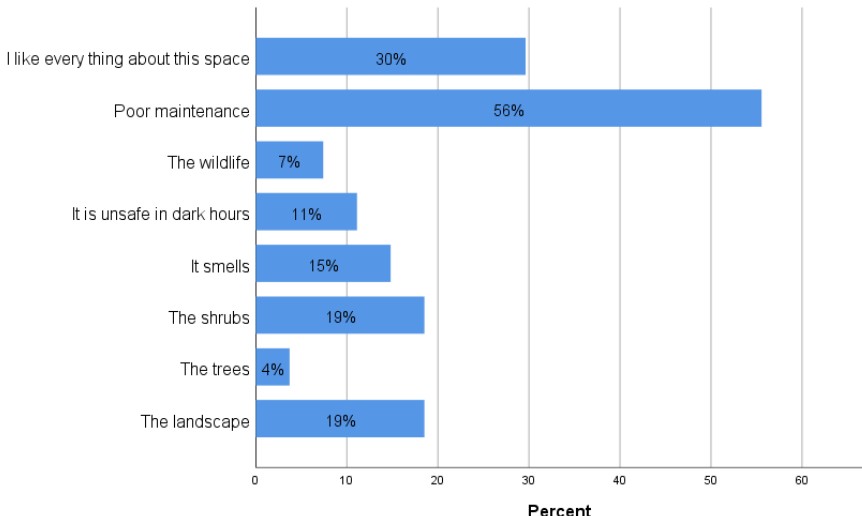

**Figure 7.** Respondent views on what they do not like about their local bioretention basin.

Some of the free text responses also revealed concerns around the safety and design of bioretention basins. As noted by one respondent, '*This area is specifically a family neighbour-hood and with so many young children and pets it needs to be made safer*'. Other respondents felt that bioretention basins are areas with greater potential—well beyond their designed purpose. One resident commented that '*More functional design of these areas so they can be used for recreation and education.*' The sentiment that multiple benefits, including but also beyond stormwater management, could be achieved through re-design was highlighted by statements like '*I support the need for the water filtration system and clean runoff. Would prefer to keep the feature but some redesign preferred.*'

Resident's views were explored more deeply in the survey after respondents were provided with a detailed explanation of how bioretention basins are designed and function to achieve improved water quality in urban streams (see Supplementary Materials). After being informed about the function of the feature, there was an 11% increase in the degree to which respondents had a more positive view on the feature. Indeed, more than 75% of respondents indicated that the bioretention basin enhanced their local environment. Furthermore, the difference between the group that had originally identified the primary function and those that had not was markedly reduced (71% vs. 59%).

*3.6. Residents Views on Council Performance on Waterways Health*

The final question in the survey explored the overall level of public satisfaction with the local council's management of stormwater. This question was important for the local council to try and understand their performance. Around half (47%) expressed the view that the local council was doing enough to ensure healthy waterways by treating stormwater (Figure 8). However, a similarly large number neither agreed nor disagreed. The results suggest that many people are either unaware of the efforts that have been taken or have no opinion on them. Importantly, those that are aware have a generally positive view.

The free text responses highlighted some of the positive council actions recognised by the participants. Many residents had positive things to say about the work of the local council in protecting stormwater. As one resident commented '*Council is working hard to improve stormwater quality, associated with new development - this includes some monitoring of outcomes post construction.*' However, the challenges for councils were also highlighted. One resident noted the challenge of understanding whether or not the design objectives were

being met with their observation that '*More council resources required to monitor outcomes*', as well as the more specific maintenance and design challenges discussed above.

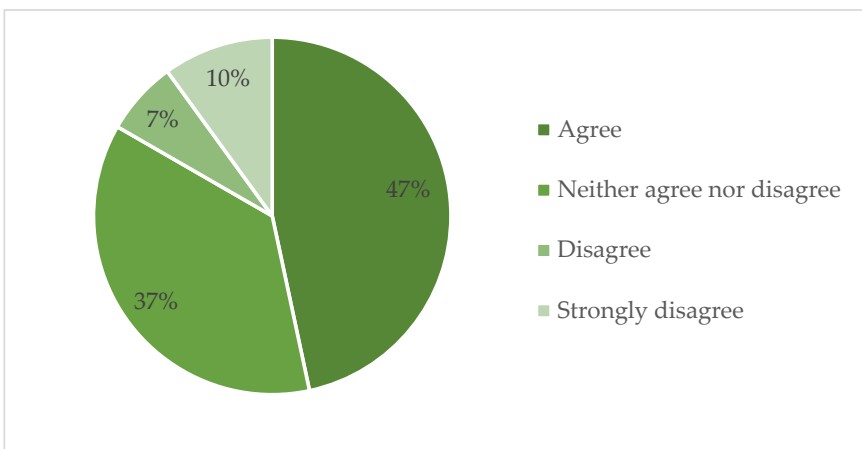

**Figure 8.** Respondent views regarding their agreement or disagreement with the statement "My local council is doing enough to ensure healthy waterways by treating stormwater."

## 4. Discussion

This pilot study survey has revealed that most residents do not know what their local bioretention basins have been designed to achieve, but they have attached additional values to these systems in terms of their contribution to green space and provision of habitat for wildlife. This limited knowledge, even among these engaged survey respondents, aligns with other results of community knowledge of water management more widely and WSUD is particular [46,48]. Water management and WSUD in Australia have become highly technocratic and focused on engineering solutions, with less focus on community engagement [40,57–59]. This is despite the growing body of research that has drawn attention to the benefits of engaging communities in Australia and elsewhere [60–63]. Engaged communities are seen as a key component to 'water sensitive cities' [9,12,64] and WSUD [8,28,45].

Education and awareness is critical to engaging communities securing broad community acceptance of stormwater treatment assets like bioretention basins, particularly in relation to their specific design objectives to store and treat stormwater prior to discharge into local streams. Residents recognise that these features are 'designed' and the results here suggest that when informed of the details and purpose of the design, they do value these outcomes. Clearly, residents need to be made more aware of the actions and infrastructure in place to achieve these goals. Indeed, this study showed that once respondents were aware of the design and purpose of the bioretention basins they revealed stronger interest in these systems and appreciation of the additional values they might bring.

A first step to raising resident awareness is probably something simple, such as installing signage. In fact, as a result of this study Brisbane City Council is considering installing information signs on stormwater features across the city. Such signage will raise awareness of both waterway health and the actions being taken to protect it. When signs are placed in close proximity to the WSUD assets, residents will become more aware of the feature and will begin to build some connection to it. Of particular relevance to our study participants, well-crafted signage will also increase the technical knowledge of more highly engaged residents who have an interest in the details. This is only one small part of engaging communities, and is likely to only reach a limited number of people, but it is a relatively low-cost and straight-forward way to begin engagement. Also, from a local council's point of view, it advertises the action they are taking and how money is being spent to improve the environment, which is often not recognised by rate-payers.

The fact that residents have attached other values to these spaces provides important opportunities both for further engagement and to improve the design of features to raise the

levels of acceptance even higher. The limited knowledge of residents around stormwater quality and management mean they are more interested in other environmental values of these features. Tapping into these multiple benefits could provide an opportunity to engage with communities more broadly around catchment and environmental stewardship, as well as increase acceptance of WSUD features. The importance of multiple benefits for WSUD and green infrastructure has been widely discussed [24,33,65–67]. However, the case of stormwater treatment assets in this study highlights the challenges, as the assets are often installed by the government or by developers with the aim of meeting a State-level planning policy on water quality, and so the water quality function is the main focus, with other functions given less (or no) importance. While Brisbane City Council has sought to encourage developers to design WSUD features by providing a route to develop context-specific assessments within the development assessment process, most new WSUD assets have tended to be based on standard designs, with a view to meeting the planning objectives. With more incentives, particularly linking WSUD assets to community values and, by extension, house values, there is scope for Councils to further encourage developers to adopt designs that go beyond the legislated stormwater management guidelines.

Of course, adding extra functions to these features while maintaining the expected water quality performance may not be straightforward. For example, some of the stormwater features surveyed in this project are not publicly accessible and are fenced off to ensure their performance is not affected by public use, thus, preventing some other use functions. The design of the stormwater asset may limit the conservation uses as well, although ensuring native species are used and preventing public access may boost conservation uses.

Nonetheless, having multiple functions of the stormwater treatment assets provides a greater business case for the continued implementation and management of these features within governments. Increased awareness and non-market valuation of bioretention basins by governments and residents could offer opportunities for council to expand their view of the value of these systems, and, in turn, establish mechanisms to provide more funds for the ongoing maintenance of bioretention basins owing not just to their design objectives, but also to the additional values that residents have attached to them [68].

The limited response rate of this pilot survey means the results must be considered in context. The demographics of respondents suggest a disproportionate response from highly educated socio-economic groups and our respondents are more likely to be people that have an interest in either WSUD or greenspace more generally. We suggest therefore, that the survey likely over-estimates the knowledge of the community about the function of the spaces, and that the need of awareness and education is probably far greater than this pilot suggests. However, despite the relatively low response numbers, we suggest that this survey is likely a good reflection of perceptions of people who are engaged—those that use and have an attachment to the public areas that contain the stormwater assets. Furthermore, these responses likely represent the views of groups who might be considered the easiest to engage in future awareness and education efforts.

Importantly, the pilot project has also revealed a number of challenges to accessing community perceptions of WSUD features through a survey. Most importantly was the challenge of getting sufficient responses. A survey of this nature requires people to recognise a photograph of the feature and each feature is likely to have a limited 'catchment' of people that will recognise it. Our pilot did try a number of methods, including on-site intercept surveys, online social media recruitment and incentives. Notably, the incentive did not seem to increase participation. Online sharing of the survey was reasonably successful as well, and given more resources, it is possible that targeted advertising could be used. Our onsite intercept surveys had limited effectiveness, but the project did not have sufficient resources to spend long periods of time, consistently at each site. We suggest that with greater resourcing, a greater response rate could be achieved.

The limited number of responses in this pilot meant that the survey potentially suffered from the same limitation as other methods (interviews or focus groups)—that only those already somewhat engaged were willing to take part. Generally, a survey method is used to

try and investigate the views of a greater cross-section of a group (here the general public). Nonetheless, given the lack of data on community perceptions, the study demonstrates that useful and important results can be obtained about public perceptions of WSUD. The results show that with careful design and avoiding the use of jargon, the public can be meaningfully asked about highly technical WSUD features and functions. Importantly, in this case, the survey also provided a way to educate the public about the use of the features.

## 5. Conclusions

This study has proposed and initially tested a method for investigating community perceptions of stormwater assets. It highlights and seeks to address the challenges of gathering data about very local and potentially unrecognised WSUD features. Notwithstanding the challenges of getting responses, the results suggest the method is viable and could be used on a wider scale by researchers or local governments seeking to understand community perceptions and acceptance of WSUD features.

Resident feedback revealed interest, at least for the engaged participants surveyed here, in bioretention basins and the efforts of Council to improve local environments and, by extension, the liveability of neighbourhoods. It is clear from the survey results that residents largely support the implementation of bioretention basins and their waterway health outcomes. Furthermore, with some design modifications, revised maintenance and improved awareness campaigns, it is likely that support will continue to grow for these stormwater management devices as the additional benefits attributed to these features by local residents are realised.

This work was specifically designed and conducted to support local councils, who face decisions over spending, design and siting of WSUD features but often lack data about public acceptance. Our exploration of community perceptions of the values of bioretention basins provide clear indications of where more work is needed to both provide useful information to residents and to get feedback from residents' views to Council and designers of stormwater infrastructure. Our recommendations to Council, to the stormwater asset design industry and to researchers, are as follows:

On the basis of this pilot study we suggest that:

- Education and awareness has a vital role to play around acceptance and valuing stormwater treatment assets with respect to their designed functions.
- Community-wide knowledge of stormwater treatment assets may help to encourage broader catchment stewardship in communities.
- Increased community awareness can support valuation of stormwater treatment assets across multiple benefits, which, in turn, could provide justification for funds for maintenance and retrofitting schedules.

We recommend that councils consider signage and education programs to grow community and resident support for the maintenance of stormwater treatment assets and develop funding mechanisms to facilitate their upkeep. The results also suggest that the Stormwater Design Industry increase efforts, where practicable, to include multiple benefits in new designs for stormwater treatment assets, and especially those that respond to community values (social design) in new designs for stormwater treatment assets.

Finally, we suggest there is further work for researchers in this area, including:

- More closely exploring community perceptions and values of more diverse stormwater treatment assets.
- Investigating new ways to capture the views of those not already engaged in some level, possibly including new ways of targeting respondents to increase response rates.

On the basis of the abovementioned recommendations, it is clear that more work in this space will enable the enhancement and ongoing support for the rapidly-expanding stormwater asset portfolios in local councils throughout Australia.

**Supplementary Materials:** The following are available online at https://www.mdpi.com/2413-885
1/5/1/5/s1, Figure S1: Example copy of the survey used in this research.

**Author Contributions:** Conceptualization, E.A.M., W.L.H. and H.Z.; methodology, H.Z., E.A.M.
and W.L.H.; software, H.Z.; formal analysis, H.Z., E.A.M. and W.L.H.; investigation, H.Z., E.A.M.
and W.L.H.; writing—original draft preparation, E.A.M., W.L.H. and H.Z.; writing—review and
editing, E.A.M. and W.L.H.; visualization, H.Z. All authors have read and agreed to the published
version of the manuscript.

**Funding:** This research received no external funding.

**Institutional Review Board Statement:** The study was conducted according to the guidelines of the
Declaration of Helsinki, and approved by Ethics Committee of Griffith University (protocol code
2018/841 19 October 2018.

**Informed Consent Statement:** Informed consent was obtained from all subjects involved in the study.

**Data Availability Statement:** The data presented in this study are available on request from the
corresponding author. The data are not publicly available due to ethical restrictions on the use of
the data.

**Acknowledgments:** The authors would like to thank the financial support from the Cities Research
Institute, Griffith University

**Conflicts of Interest:** The authors declare no conflict of interest.

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
