# Peer review of "Community Perceptions and Knowledge of Modern Stormwater Treatment Assets"

_urbansci, doi:10.3390/urbansci5010005_

Round 1

Reviewer 1 Report

The manuscript by Zamanifard et al., investigated the perceptions of society about stormwater treatment measures in Brisbane with a surveying approach. Overall the work that has been done shows an extensive effort that deserves appreciation. The study concludes that there are some limits in the knowledge of society about WSUD treatment measures which are very useful notice for the local governments. The submitted manuscript merits a publication and can be published in the urban science journal after doing some edits and adding more results:

The authors showed most of the results in one-way figures. I suggest authors to also conduct an analysis to show the interactions between question choices. A simple correlogram (with considering percentages or frequencies of common answers)/bubble plot/ or heat map between different question choices can help the analysis to show if there is any interaction between questions and answers of the people who participated in the survey. For example, it would be interesting to see the interactions between answers of what people like and dislike about WSUD treatment measures. and many other questions like the relation between age and different answers, and etc.

Apart from this the paper looks fine and needs just minor corrections in the text:

Line 36-38: (In respect to stormwater...local streams and rivers) Please re-write this sentence.

Line 140: Table 2: it is Table 1.

Line 140: Table 2: Please write in the caption what "Brisbane (%)" means.

Author Response

Reviewer comment: The manuscript by Zamanifard et al., investigated the perceptions of society about stormwater treatment measures in Brisbane with a surveying approach. Overall the work that has been done shows an extensive effort that deserves appreciation. The study concludes that there are some limits in the knowledge of society about WSUD treatment measures which are very useful notice for the local governments. The submitted manuscript merits a publication and can be published in the urban science journal after doing some edits and adding more results:

Response: We thank Reviewer #1 for these supportive and positive comments on our paper. We agree that local governments will find this work particularly interesting as they become responsible for more of these stormwater treatment assets.

Reviewer comment: The authors showed most of the results in one-way figures. I suggest authors to also conduct an analysis to show the interactions between question choices. A simple correlogram (with considering percentages or frequencies of common answers)/bubble plot/ or heat map between different question choices can help the analysis to show if there is any interaction between questions and answers of the people who participated in the survey. For example, it would be interesting to see the interactions between answers of what people like and dislike about WSUD treatment measures. and many other questions like the relation between age and different answers, and etc.

Response: We thank the reviewer for these analytical suggestions and we did explore relationships in responses to questions through a variety of approaches in our initial analysis. In particular, we sought to explore if there were any differences in response data from the group of respondents that knew what the stormwater treatment assets were, against the remainder of the sample pool who did not know what the stormwater treatment assets were. We felt this would be the most interesting difference given the scope of the paper. The only significant association was for responses for two very similar questions – namely “How desirable is it to have this green space in your vicinity?” and “Do you think this feature enhances or diminishes your local environment?”. Unsurprisingly, respondents that felt that the space was desirable, also felt that it enhanced the local environment. As this finding is not particularly noteworthy, we have not included these analyses in the revised manuscript. Since no other comparisons (including demographics) were significant, we don’t provide these in the revised manuscript either.

We acknowledge that the sample size of this study makes more sophisticated quantitative analyses impossible. This is in part due to the different perceptions across the range of assets explored as well as the fact that multiple questions enabled multiple responses which means that clean, direct, associations across questions quite difficult to undertake.

We have added a short paragraph, line 217, to mention the fact that there were no significant trends. However, we have noted one interesting possible correlation: those that knew the primary function of the feature were more likely to consider the feature desirable. We have discussed this briefly, and with some caveats, in added sentences (line 251) and also returned to it briefly in line 288. We are grateful that the reviewers  comment prompted us to re-consider this and include it.

Reviewer comment: Apart from this the paper looks fine and needs just minor corrections in the text:

Line 36-38: (In respect to stormwater...local streams and rivers) Please re-write this sentence.

Response: The sentence has been re-written and now reads as follows:

“A water sensitive approach is designed to slow the flow of stormwater, to help manage flood risk and facilitate water treatment through settlement and assimilation processes, prior to discharge into local streams and rivers”

Line 140: Table 2: it is Table 1.

Response: We thank the reviewer for picking up this error. We have changed the reference to Table 1 and have also changed the Table title to reflect this change.

Line 140: Table 2: Please write in the caption what "Brisbane (%)" means.

Response: We thank the reviewer for this suggestion. We have modified the Table title to provide this additional information. The Table title now reads as follows:

“Table 1. Demographic details of survey respondents, with comparisons (where relevant) to comparable data for the greater Brisbane region (Brisbane (%)).”

Reviewer 2 Report

  • Based on my understanding the "correct" answers to the given question in Section 3.2, are considered as "stormwater treatment asset" and "natural filtration feature". I think WSUD features have multiple benefits (more than two factors that are selected as correct answers). For example, why a basin can not play a role as a flood control feature or a community garden. From the line 169-170 it can be implied that the authors disagree with the multiple benefits of the basins in urban design, although in the introduction it is noted that the WSUD features are designed to have several benefits for the societies. In my opinion, the survey questions and suggested answers are misleading. One thing that I would recommend to the authors is revising the interpretation of the surveys (perhaps there are more "correct" answers?).

  • I think addressing these items can help the audience to better understand the climate condition of the studied area.
    1. Some information about the climate and weather conditions of the studied area such as average annual precipitation, temperature, etc can be added.
    2. How often are these basins have played the role that they are designed for (stormwater infiltration and treatment)?
    3. What are the designed return periods for the studied basins?
  • Figure 4 is not crossreferenced in the manuscript. It should be mentioned and discussed.

Author Response

Reviewer comment: Based on my understanding the "correct" answers to the given question in Section 3.2, are considered as "stormwater treatment asset" and "natural filtration feature". I think WSUD features have multiple benefits (more than two factors that are selected as correct answers). For example, why a basin can not play a role as a flood control feature or a community garden. From the line 169-170 it can be implied that the authors disagree with the multiple benefits of the basins in urban design, although in the introduction it is noted that the WSUD features are designed to have several benefits for the societies. In my opinion, the survey questions and suggested answers are misleading. One thing that I would recommend to the authors is revising the interpretation of the surveys (perhaps there are more "correct" answers?).

Response: We thank the reviewer for these comments, because they highlight the need for us to be very clear in how our study was framed. The reviewer is correct – for the purposes of our study and the bioretention basins in question, we were interested in whether residents understood that these have been constructed in response to State Planning Policy (SPP) guidelines around the need to treat stormwater draining from the local area. Importantly, the SPP does not stipulate the need for flood mitigation features, communal gardens or any of the other attributes that may be associated with these bioretention basins. As the reviewer notes, there may be multiple benefits – and indeed, this is what our study has been designed to uncover – the wide range of community perceived benefits of stormwater treatment assets and the value that local governments can derive from further communication with residents around the designs and purpose of these systems.

The reviewer’s assertion that we disagree with the multiple benefits is therefore incorrect – we were simply linking our survey to the legislated design objectives for these systems. Indeed, we absolutely agree that these systems have many more benefits and we believe that the findings of this study represent an important step towards recognising multiple benefits in the eyes of the residents that live near to these stormwater treatment assets.

With all of the above in mind, we have re-written some of the text in the Introduction to re-emphasise the design intent of the survey and the specific context in which we are operating, in terms of canvassing views on assets that have been specifically designed to treat stormwater. The re-written text between lines 50-60 as follows:

“WSUD features have become increasingly part of water management and planning policy in rapidly developing urban areas [10,11,17]. For example, in the area where this study is conducted, Queensland, Australia, WSUD is now a legislated requirement for all new developments. This means that new urban developments must have a stormwater management plan and adopt features of WSUD to treat stormwater onsite. The state planning policy has set performance outcomes and design objectives for nutrient removal via stormwater management that provide minimum reduction requirements for total suspended solids, total phosphorus, total nitrogen, and gross pollutants for WSUD stormwater assets [31]. While there are no doubt other benefits associated with stormwater treatment assets, including flood control, stormwater harvesting and biodiversity [10,16,23,26,27], the specific design objectives of interest in this study related to their role as treatment systems, which is the common focus of stormwater features in the US, Europe and Australia [16].”

 We have also changed the text in the results to make it clearer by referring to the ‘primary’ or ‘as-designed’ function when referring to the ‘correct’ answers.

Reviewer comment: I think addressing these items can help the audience to better understand the climate condition of the studied area.

1. Some information about the climate and weather conditions of the studied area such as average annual precipitation, temperature, etc can be added.

Response: We have included some climate information for the region,lines 99-113,  which will help readers understand the local context.

2. How often are these basins have played the role that they are designed for (stormwater infiltration and treatment)?

Response: We thank the reviewer for this question – it is one that we have pondered a lot ourselves. At present there is little to no monitoring of the performance of these systems, even including how often they play the role they are designed for. Given the climate in Brisbane (as noted in the additions mentioned above), they likely play a role relatively frequently, especially through summer, but it is in fact impossible to say. Currently, their performance is assumed based on models that inform their designs and this remains a major limitation within this field. The relevant text regarding this point lies between lines 61-66 in the Introduction, and reads as follows:

“Whilst the rationale and legislation around WSUD is compelling, there remain many significant knowledge gaps around the performance and environmental outcomes of WSUD features post-implementation [6,19,22,26,32–36]. Indeed, WSUD performance is generally not monitored at all and expectations on sediment and nutrient reductions, or other performance measures, are generally taken as modelled, without the necessary ground-truthing, although more recent studies have begun to investigate in-situ performance [22,23,33,35,37]..”

3. What are the designed return periods for the studied basins?

Response: We thank the reviewer for this comment, but believe that these aspects of the designs lies beyond the scope of the current study. Indeed, the design specifications are determined on the basis of the climate – which is information we now provide in this paper – and the topography and size of the catchment being treated. There are a wide range of design standards and these include overflow paths during very large storm events. In simple terms, all of the bioretention basins our respondents were familiar with are designed to treat all but the largest rainfall events. As mentioned above, the degree to which these returns are realised and performance standards are met, remains a significant knowledge gap.

4. Figure 4 is not crossreferenced in the manuscript. It should be mentioned and discussed.

Response: We thank the reviewer for pointing this out – Figure 4 is now crossreferenced in the manuscript text.

Reviewer 3 Report

Some findings of this manuscript are good. However, the respondent number is too low (n=31) for an online survey. I encourage the authors to collect data from 100 or more respondents.

Literature review is also little old. Articles published from 2017 are not properly utilized.

Author Response

Reviewer comment: Some findings of this manuscript are good. However, the respondent number is too low (n=31) for an online survey. I encourage the authors to collect data from 100 or more respondents.

Response: We thank the reviewer for this comment. We agree that ideally more respondents is always better. However, we believe that the consistent opinions offered across our respondent pool reflects a reasonable and representative sub-sample of the population that live within the vicinity of the bioretention systems. One of the challenges associated with getting larger respondents numbers lies in the fact that each survey for each location relies on residents to recognise the system (and be willing to reply to the survey). We have noted this and other limitations in our Discussion. Nevertheless, we do feel that the current sample size is sufficient to draw conclusions around resident perceptions and knowledge. We also note that the other two reviewers have not commented on sample size and felt satisfied with the results and findings of the study.

Reviewer comment: Literature review is also little old. Articles published from 2017 are not properly utilized

Response: We thank the reviewer for this comment and have updated our reference list with some more recent publications, particularly in the Introduction. We note that some of the older references are particularly relevant for this context and these issues, but we have now better acknowledged that there has been more recent work in this space more generally. However, the specific issue we are discussing – public perceptions – has limited work in this space as it remains a growing field of research activity.

Round 2

Reviewer 1 Report

The revised version of the manuscript by Zamanifard et al., have some modification in the text in order to cover my previous comments. However, I was expecting the authors to do multidimensional analysis to see if meaningful outcomes have been driven out of surveying or not? I think that (as also the authors claim), the number of surveys (i.e., 31) do not allow the authors to expand their statistical analysis. It seems that the authors need to increase their number of surveys (maybe at least 100) in order to support their results. I was hoping to see the outcomes of multidimensional analysis in-line with current results, while authors also claim that answers from different questions do not support each other. Hence I suggest the authors to apply these major revisions (particularly to increase the number of surveys) and re-submit their manuscript for evalutation.

Author Response

Response: We thank Reviewer 1 for these comments. The suggestion of a multi-dimensional analysis is an important one. As we note in the revised version of the manuscript, we have conducted an initial multidimensional analysis, investigating trends between sets of questions (e.g. what people thought it was used for and what they liked/disliked about it). However, as we note in the revised paper (line 219-222), there were no statistically significant trends emerging from this analysis. We did, however, include in the text the strongest correlation between those who understood the designed function and those who did not – those that knew what the feature was designed for were more likely to consider it desirable (line 253-258); although again this result was not statistically significant (Spearman’s rank test).

Of particular importance for this manuscript, we believe that the types of multidimensional statistical analyses being suggested here are limited by two factors:

(1) The types of questions being asked and the design of the survey in order to investigate resident perceptions. The survey design was led by the research questions and resulted in a mixed methods survey approach (coupled quantitative and qualitative data) which sought out the views and perceptions of residents. These design features make statistical analysis complicated - for example, in several questions respondents were allowed to choose more than one response to allow us to uncover the likely plurality of views and perceptions. As a result, analysing multidimensional correlations between people’s views e.g. on function and what they liked or disliked is not straightforward. We do not see this as a weakness of the work; indeed, this investigation of pluralities of views and perceptions is key to the paper and this is why we do not rely solely on the quantitative data and statistical analyses. Instead, we learn more about perceptions from the qualitative, free-text responses of residents and these are more important to our findings given that perceptions can rarely (if ever) be described with quantitative data alone. In summary, this mixed methods approach may limit the scope for statistical analysis, such as those being suggested by Reviewer 1 here, but it opens up a deeper analysis of perceptions and can provide more critically useful insights for researchers and local government practitioners who are responsible for the maintenance of bioretention basins in their jurisdictions.

(2) the number of respondents – we acknowledge, as we did in the last round of reviews, that more respondents would, of course, be better. However, as discussed in the paper, this is also not a straightforward matter of simply releasing the survey again. The number of respondents is inherently limited due the spatial nature of the features being studied – there will only ever be a limited number of people aware of a single feature in their locality. We discuss this issue and the associated challenges with getting high response rates, in detail in the paper. In the context of the current manuscript, achieving the large sample size suggested by the reviewer would require substantial further work, the accompanying resources, as well as likely working closely with government partners to promote and support the survey. We also are concerned that more respondents would sway us towards more quantitative analysis and would make it more difficult to tease apart the rich qualitative data which we also wish to highlight in this manuscript.

In summary, we believe that the manuscript, reporting the results of this pilot study and highlighting the possibilities and challenges of this type of investigation, represents an important and significant contribution to limited current literature examining resident perceptions of stormwater features.

Finally, we disagree with the reviewer’s comment that the “authors also claim that answers from different questions do not support each other”. Indeed, we highlight throughout the paper that the answers from different questions do tend to support each other. For example, those that found the features desirable also thought that the feature enhanced the local environment. This is not surprising, but was also an important part of the study - making sure that there was consistency across questions and to check that people were answering thoughtfully. Furthermore, we also highlight how the qualitative data, coupled with the quantitative data, enables a richer exploration of the thinking behind respondent views and can increase our confidence in our interpretations of the presented findings.

Reviewer 3 Report

I am almost OK with the revisions provided by the authors. This manuscript now could be published in the journal.

Thanks for considering me in the reviewing process of this manuscript.

Author Response

Response: We thank Reviewer 2 for this very positive response and for recommending publication of the manuscript in this journal. We do believe that it represents an important and highly citable contribution to this field of research.

Round 3

Reviewer 1 Report

The revised version of the manuscript by Zamanifard et al. has tried to cover my comments. However, still I think that this study needs a certain justification. During the past days, I was doing a literature review to see if there are similar studies (with very few survey observations) and unfortunately, I could not find other studies to support the submitted manuscript.

The authors need to add a discussion in their manuscript and support their decision (conducting research with limited survey observations; eg: less than 50) by including some examples of other studies (particularly up-to-date ones) that conducted surveying type research and reached to solid conclusions with limited observations.

Otherwise, I think that the paper cannot be accepted for publication with the current format.